# Spontaneous Improvement of Aortitis Associated with Severe COVID-19 Infection—A Case Report

**DOI:** 10.3390/medicina59050816

**Published:** 2023-04-22

**Authors:** Takashi Shimada, Hideya Itagaki, Yuko Shirota, Tomoyuki Endo

**Affiliations:** 1Division of Emergency and Disaster Medicine, Tohoku Medical and Pharmaceutical University Hospital, 1-15-1, Fukumuro, Miyaginoku, Sendai City 983-8512, Miyagi, Japan; 2Division of Hematology and Rheumatology, Tohoku Medical and Pharmaceutical University Hospital, 1-15-1, Fukumuro, Miyaginoku, Sendai City 983-8512, Miyagi, Japan

**Keywords:** aortitis, COVID-19, large vessel vasculitis, SARS-CoV-2, leukocytoclastic vasculitis

## Abstract

Aortitis is a rare complication of the coronavirus disease 2019 (COVID-19) and is often treated empirically with steroids. We present a case of spontaneous resolution of aortitis without treatment. A 65-year-old man was admitted to our intensive care unit for severe COVID-19 pneumonia and underwent rehabilitation in the general ward. On day 12, he developed fever, and on day 13, he developed right cervical pain and increased inflammatory markers. On day 16, a cervical echocardiogram showed vasculitis in the right common carotid artery, and on day 17, computed tomography (CT) of the neck showed thickening of the arterial wall of the right common to the internal carotid arteries. A retrospective assessment of the CT scan on day 12 showed wall thickening from the thoracic aorta to the abdominal aorta, and a diagnosis of aortitis was made. Autoantibody analysis, culture, and magnetic resonance imaging (MRI) of the head and neck showed no abnormalities. During the investigation of the cause of aortitis, the fever and inflammatory reaction spontaneously resolved and the right cervical pain gradually improved. Therefore, the patient was diagnosed with transient COVID-19-related aortitis. To our knowledge, this is the first report describing the spontaneous resolution of COVID-19-related aortitis.

## 1. Introduction

Coronavirus disease 2019 (COVID-19) is a viral infection caused by severe acute respiratory syndrome coronavirus 2 (SARS-CoV-2) that was first reported in Wuhan, China on 31 December 2019, and subsequently became a global pandemic. Common symptoms of COVID-19 include cough (50%), fever (43%), myalgia (36%), headache (34%), and respiratory distress (28.5%) [1]. Other reported complications include taste and smell disorders, myocarditis, encephalitis, and vascular events such as myocardial infarction, pulmonary embolism, and cerebral infarction [1,2]. Although rare, vasculitis and aortitis have also been reported to be complications of COVID-19 and are usually treated with steroids. In the present case, three weeks after the COVID-19 infection, the patient developed aortitis with fever and neck pain that resolved spontaneously without any specific drug treatment. To the best of our knowledge, no case of spontaneously resolving COVID-19-associated aortitis has been reported to date, and this is the first report describing such a case.

## 2. Case Presentation

A 65-year-old Japanese man was brought to our hospital with severe dyspnea. Nine days before visiting our hospital, he had developed a sore throat, fever, and cough. He tested positive for the COVID-19 antigen at home and was prescribed an antipyretic analgesic (acetaminophen) and expectorant (ambroxol/l-carbocysteine) by his family clinic the next day. Six days prior to his visit, his dry coughing symptoms worsened, and headache and dyspnea also appeared; he was admitted to another hospital one day before visiting our hospital. The oxygen saturation (SpO_2_) was 93% (room air) on admission, but soon after admission, it dropped to the 80% range even with an oxygen mask at 5 L/min. He was switched to a high-flow nasal cannula (HFNC), and the SpO_2_ increased to 90–92% at a flow rate of 40 L/min and a fraction of inspired oxygen (FiO_2_) value of 60%. The blood gas analysis showed the following findings: pH, 7.46; partial pressure of oxygen (PO_2_), 78.8; and partial pressure of carbon dioxide (PCO_2_), 36.6. The treatment included remdesivir 200 mg/day and methylprednisolone 1000 mg/day for COVID-19 pneumonia, tazobactam/piperacillin 4.5 g three times/day for bacterial infection complications, and heparin 15,000 units/day for thromboprophylaxis. However, the previous physician was unable to cope with the rapid worsening of the patient’s respiratory condition, and he was transferred to the intensive care unit of our hospital.

The patient’s medical history included hypertension, and he was taking amlodipine (5 mg). His vital signs on arrival were as follows: blood pressure, 177/92 mmHg; pulse, 97 bpm; respiratory rate, 25 breaths/min; SpO_2_, 93–95% (HFNC; flow: 60 L/min and FiO_2_: 60%); and body temperature, 36.8 °C. Arterial blood gas analysis showed the following findings: PaO_2_, 81.8 mmHg; PCO_2_, 34.6 mmHg; HCO^3−^, 23.2 mmol/L; lactate level, 3.6 mmol/L; and P/F, 136. Blood tests revealed slightly elevated levels of liver enzymes (aspartate aminotransferase, 70 U/L; alanine transaminase, 49 U/L; and lactate dehydrogenase, 487 U/L), increased inflammatory response (C-reactive protein (CRP), 188 mg/L), and impaired coagulation (activated partial thromboplastin time > 200 s; prothrombin time/international normalized ratio, 1.4; antithrombin III, 68%; and D-dimer, 1.88 µg/mL) (Table 1). Chest computed tomography (CT) showed a bilateral diffuse ground-glass appearance with infiltrative shadows just below the pleura (Figure 1). The patient was treated with remdesivir, dexamethasone (for 10 days), tazobactam/piperacillin, and heparin. On day seven of hospitalization, he was transferred from the intensive care unit to the general ward for rehabilitation. 

However, on day 12 of hospitalization, the patient developed a fever of 38.1 °C. To investigate the source of the fever, blood tests, urinalysis, blood culture, and contrast-enhanced CT scans of the neck, thorax, abdomen, and pelvis were performed. Blood tests showed a white blood cell (WBC) count of 14.4 × 10^3^/μL (neutrophils: 79.0%, lymphocytes: 4.0%, monocytes: 17.0%, basophils: 0%, and eosinophils: 0%) and a CRP level of 290 mg/L, while the urinalysis showed positive results for white blood cells and nitrite. Contrast-enhanced CT revealed no abnormalities in the renal parenchyma. However, based on the urinalysis and elevated inflammatory response, we considered the possibility of a urinary tract infection and started 500 mg levofloxacin on the same day. However, the antimicrobial agents did not relieve the fever, and the right cervical pain newly appeared on day 13 of hospitalization. A central venous catheter was inserted through the right internal jugular vein from the day of admission to day eight; we considered the possibility of infectious thrombophlebitis and performed neck echocardiography on day sixteen of hospitalization. A neck ultrasound showed a hypoechoic area (Figure 2) surrounding the right common carotid artery, which was suspected to indicate vasculitis. On day 17 of hospitalization, contrast-enhanced CT of the neck showed wall thickening and increased periprosthetic fatty tissue density in the right common, internal, and external carotid arteries, especially in the right internal carotid artery, with narrowing of the lumen (Figure 3). 

Based on these results, contrast-enhanced CT was performed on day 12 of hospitalization, and wall thickening (Figure 4) was observed from the thoracic aorta to the abdominal aorta. Aortitis was diagnosed based on these results, and a close examination of the cause was conducted. Immunological tests showed negative results for rheumatoid factor, anti-nuclear antibody, whole anti-Sm antibody, anti-RNP antibody, anti-ds DNA antibody, anti-SS-A body, anti-SS-B body, anti-neutrophil cytoplasmic antibody (ANCA), whole MPO-ANCA and C-ANCA, and anti-cardiolipin IgG antibody. The complement levels were elevated (C3, 1910 mg/L; C4, 390 mg/L; and CH50, >600 U/L). The blood culture, syphilis test, tests for hepatitis B and C, HIV test, parvovirus, interferon-gamma release assays (IGRA), and tests for other infectious vasculitis all showed negative results. The procalcitonin level was also not elevated (0.04 ng/mL). Contrast-enhanced MRI of the head and neck showed no obvious wall thickening of the temporal artery, and the cerebral arteries were well-defined. Based on these results, the cause of aortitis was considered to be COVID-19. 

The CRP level gradually decreased from day 16 of hospitalization, reaching 86 mg/L on day 20 of hospitalization. The fever resolved accordingly, and the right cervical pain also improved. No steroids or nonsteroidal anti-inflammatory drugs (NSAIDs) were administered during this period. The patient’s condition continued to improve and he was discharged on day 21 of hospitalization. An outpatient assessment performed 2 days after discharge showed that the right cervical pain had completely disappeared, and the CRP level had decreased to 32 mg/L. Based on these findings, a diagnosis of transient aortitis due to COVID-19 was made. When the patient was discharged from the hospital 14 days later, his CRP level had further decreased to 9 mg/L, and he showed no recurrence of neck pain. Ophthalmological examination on the same day revealed no findings suggestive of giant cell arteritis.

## 3. Discussion

We encountered a case of aortitis that was diagnosed due to the appearance of fever and neck pain 3 weeks after the onset of COVID-19. To the best of our knowledge, this is the first case describing the spontaneous resolution of COVID-19-associated aortitis.

Aortitis is a pathological term for all diseases that cause inflammation of the aortic wall, which can be non-infectious or infectious in origin [3,4]. The most common non-infectious causes are Takayasu arteritis (TKA) and giant cell arteritis (GCA) [3,4]. Infectious causes include hepatitis viruses (mainly types B and C), parvovirus, syphilis, tuberculosis, Gram-positive cocci (*Staphylococcus aureus*, *Enterococcus faecalis*, *Streptococcus pneumoniae*), and Gram-negative rods (*Salmonella* spp.) [3,4]. In the present case, these infectious diseases were ruled out because the infectious disease tests (blood culture, syphilis test, IGRA, hepatitis B, hepatitis C, and parvovirus) showed negative results. Based on the course of the disease, we considered aortitis due to COVID-19, but it was necessary to differentiate between TKA and GCA, both of which are distinguished on the basis of the widely used classification criteria developed by the American College of Rheumatology (ACR) [5,6]. The ACR classification criteria for TKA are as follows: (1) age of onset before 40 years, (2) claudication of the extremities, (3) brachial artery pulsatility, (4) difference in brachial blood pressure > 10 mmHg, (5) vascular murmur in the subclavian artery or aortic arch, and (6) angiographic evidence of stenosis or obstruction in the aorta, its branches, or the large arteries in the limbs [5]. When three or more of the above six criteria are met, the patient is diagnosed with TKA. However, in the present case, the patient did not fulfill the four criteria: age > 40 years, no claudication of the extremities, no weakening of the brachial artery, and no vascular murmur, so the diagnosis of TKA was not made. On the other hand, the ACR classification criteria for GCA are as follows: (1) age > 50 years, (2) new-onset headache, (3) tenderness in the temporal artery or decreased pulse unrelated to atherosclerosis, (4) erythrocyte sedimentation rate > 50 mm/h, and (5) vasculitis with mononuclear cell infiltration, granulomatous inflammation, or multinucleated giant cells in a biopsy specimen of the shallow temporal artery [6]. When three or more of the five criteria are met, GCA is diagnosed. In the present case, only two criteria could be ruled out: the absence of new-onset headache and the absence of tenderness in the temporal artery. However, the MRI showed no arteritis in the temporal artery. Therefore, typical “cranial” GCA was ruled out. GCA includes “cranial” GCA and “large-vessel” GCA [7], but “large-vessel” GCA was not excluded. However, GCA requires steroid therapy, and spontaneous improvement and no recurrence after discharge from the hospital do not fit the course of GCA. Based on these findings, we diagnosed aortitis caused by COVID-19.

The mechanism underlying COVID-19-induced aortitis is not yet fully understood. However, Bellosta et al. [8] described the following pathophysiology of thrombosis in small- to medium-sized arteries in COVID-19 patients. First, endothelial cells are infiltrated by virions, which are then infiltrated by neutrophils and mononuclear elements, leading to accelerated apoptosis and lymphoid endotheliitis and resulting in an inflammatory and thrombotic environment. In the later stages of the disease, the same inflammatory elements expand and infiltrate the same arteries, causing periarteritis, followed by leukocytoclastic vasculitis. This reaction is followed by the deposition of antigen–antibody immune complexes, IgG, IgA, and IgM, resulting in type III hypersensitivity vasculitis. These conditions can also occur in large blood vessels such as the aorta [9]. Inflammation near the heart can cause arrhythmia, as seen in cases where inflammation at the base of the aorta caused an atrioventricular block, which in turn leads to life-threatening ventricular tachycardia caused by bundle branch reentry ventricular tachycardia (BBRVT) [10]. The aortic endothelium is largely composed of angiotensin-converting enzyme 2 (ACE2) receptors, and SARS-CoV-2 is thought to invade cells via these ACE2 receptors and cause endotheliitis and leukocytoclastic vasculitis, in addition to type 3 hypersensitivity vasculitis [4].

Aortitis after COVID-19 has been previously reviewed by Hassan et al. [11], and several cases have been reported since. Seven cases of aortitis secondary to COVID-19, including our case, were included in our review (Table 2) [4,12,13,14,15,16]. The age at onset in these cases ranged from 19 to 73 years (median: 61 years), and the condition was more common in males. The symptoms varied, but the site of inflammation was associated with symptoms, e.g., abdominal pain if the site of inflammation was the infrarenal aorta. The onset of symptoms may occur at the same time as COVID-19 or two months later; therefore, it is impossible to specify a specific onset time. Since aortitis does not allow for invasive tests such as a biopsy, the diagnosis is made by CT. Diagnostic blood markers have not yet been established, but CRP, erythrocyte sedimentation rate (ESR), and interleukin-6, an inflammatory cytokine, tend to be elevated [4,17]. IgG4 is considered less relevant. Steroids (prednisolone) are often the initial treatment of choice for aortitis. Initial doses are based on the European League Against Rheumatism (EULAR) initial doses for large-vessel arteritis (GCA and TKA) and are often 0.8–1 mg/kg of prednisolone [4]. In GCA and TKA, once the disease is under control, the goal is to reduce the dose of prednisolone to 15–20 mg/day within 2–3 months, and after 1 year, to less than 5 mg/day for GCA and less than 10 mg/day for TKA [18]. However, the rate of reduction of prednisolone for aortitis caused by COVID-19 has not been determined. The only report on this topic was presented by Joao et al. [12]. Another case report described the treatment of a patient with NSAIDs instead of steroids [15]. NSAID-treated patients had a slightly longer time for symptoms or laboratory improvement than steroid-treated patients (3–14 days for steroids and 24 days for NSAIDs).

The present case involved a 65-year-old man who showed aortitis 3 weeks after COVID-19. His symptoms included fever and right neck pain, and the CT showed wall thickening of the right internal carotid artery, common carotid artery, and thoracoabdominal aorta. The findings in the right common carotid artery were particularly strong and seemed to be related to right neck pain. Blood tests revealed elevated CRP, ESR, and interleukin-6 levels. However, the case differed from previous cases since the patient received no steroids or NSAIDs. Under normal circumstances, the time to the improvement of symptoms or tests in a spontaneously resolving case would seem to be longer than that in cases treated with steroids or NSAIDs, but this case showed an improvement in 11 days, which was not different from that in steroid-treated cases (Table 2).

**Table 2 medicina-59-00816-t002:** Cases of aortitis after coronary infection, including this case.

Reference	Sex	Age	Past Medical History	Symptoms	Time from COVID-19 Onset to Dx of Aortitis	Arteritis Location	Mode of Dx	CRP (mg/L)	ESR (mm/h)	IL-6 (ng/L)	IgG4 (mg/dL)	Aortitis Rx	Time to Improvement	Duration of Rx	Outcome
Dhakal et al. [4]	M	63	PD/asthma/DM/HTN/obesity /non-ischemic cardiomyopathy	Diffuse abdominal pain, dry cough, fatigue, and generalized weakness	Concurrent with COVID19 onset	Infrarenal aorta	CT	87	57	54	187	Prednisolone 60 mg	9 days	1 month	Discharge
Mendes et al. [12]	F	19	None	General fatigue, malaise, and chest and low back pain	1 month	Descending thoracic and abdominal aorta	CT/ FDGPET	107	109	-	55	Prednisolone 60 mg	3 days	Reduce dose by 5 mg every two weeks → About 6 months	Discharge
Shergill et al. [13]	M	71	None	Extreme fatigue, poor appetite with 5 kg weight loss, and a sharp left-sided chest pain radiating to his scapula	2 months	Subclavian arteries to the iliac bifurcation	CT	185	-	28	-	Prednisolone 40 mg	2 weeks	More than 2 weeks (End period unknown)	Discharge
Zou and Vasta [14]	M	71	Cholecystectomy, rotator cuff repair	Feeling generally unwell, weight loss, and worsening thoracolumbar back pain	2 months	Aortic arch extending all the way down the aorta	CT	122	-	25	-	Prednisolone 40 mg	1 week	More than 2 weeks (End period unknown)	Discharge
Oda et al. [15]	M	71	None	Fever and productive cough	2 weeks	Abdominal aorta to the bilateral common iliac arteries	CT	177	61	-	112	NSAIDs	24 days	1 month	Discharge
Kobe et al. [16]	M	72	DM	Dyspnea and pain in the left lower leg	Concurrent with COVID19 onset	Abdominal aorta to the left common iliac artery	CT	315	-	-	-	Unknown	-	-	Death
Shimada et al. (this case)	M	65	HTN	Fever and right cervical pain	3 weeks	Aortic arch to right common carotid artery and right internal carotid artery	CT	280	63	18.3	24	None	11 days	-	Discharge

CRP, C-reactive protein; CT, computed tomography; ESR, erythrocyte sedimentation rate; FDG-PET, fluorodeoxyglucose-positron emission tomography; IL-6, interleukin 6; NSAIDs, nonsteroidal anti-inflammatory drugs; PD, Parkinson disease; DM, Diabetes mellitus; HTN, Hypertension; -; Not mentioned in the article.

## 4. Conclusions

We encountered a case of aortitis caused by severe COVID-19 pneumonia that improved without the administration of steroids or NSAIDs. However, as the initial dose of steroids and the rate of dose reduction were not recorded and some patients improve without steroids, as in this case, the accumulation of data from additional case reports is needed, and the treatment options should be carefully considered.

## Figures and Tables

**Figure 1 medicina-59-00816-f001:**
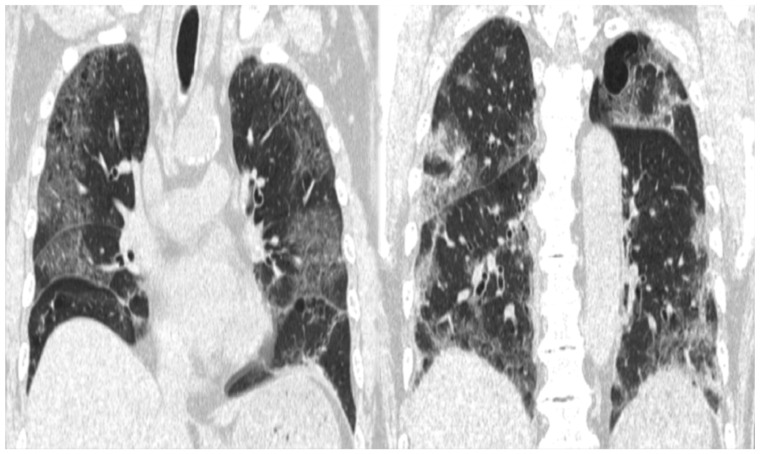
CT image showing extensive frosted shadows and sclerotic images in both lungs.

**Figure 2 medicina-59-00816-f002:**
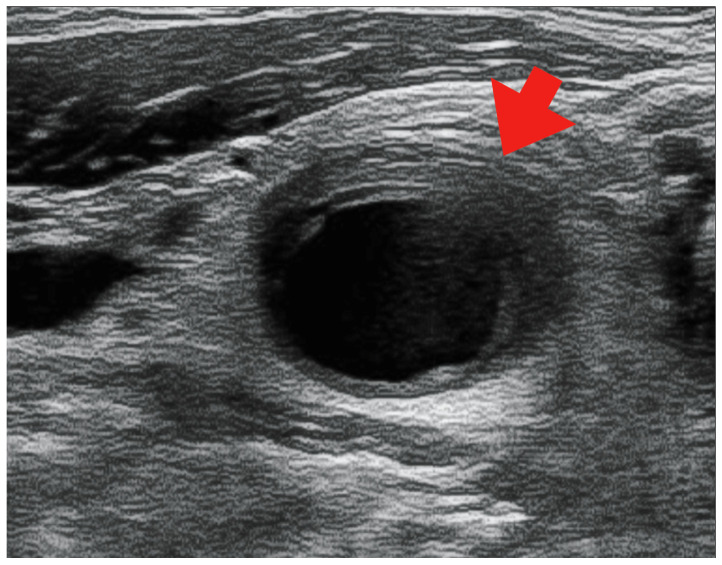
Echocardiographic image showing a hypoechoic region surrounding the right common carotid artery (red arrow).

**Figure 3 medicina-59-00816-f003:**
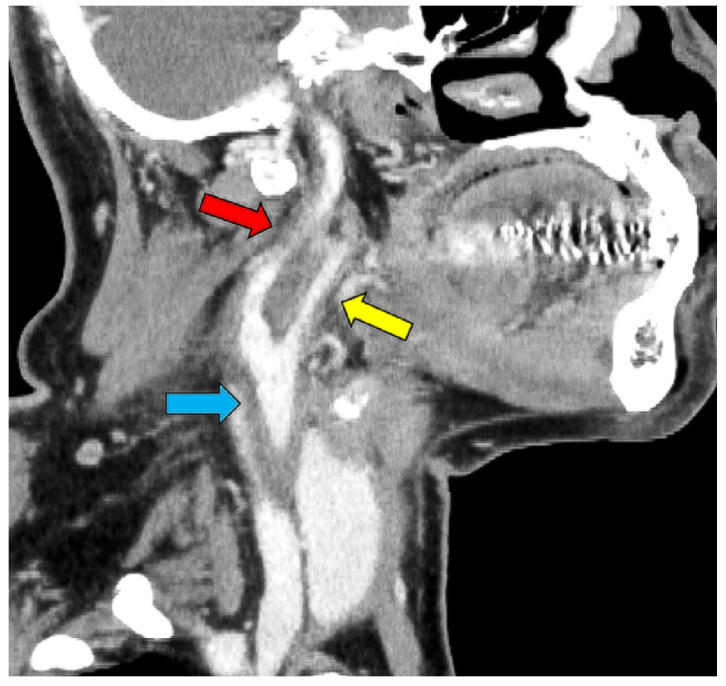
CT image showing thickening of the right internal (red arrow), external (yellow arrow), and common carotid artery (blue arrow) and narrowing of the internal carotid artery.

**Figure 4 medicina-59-00816-f004:**
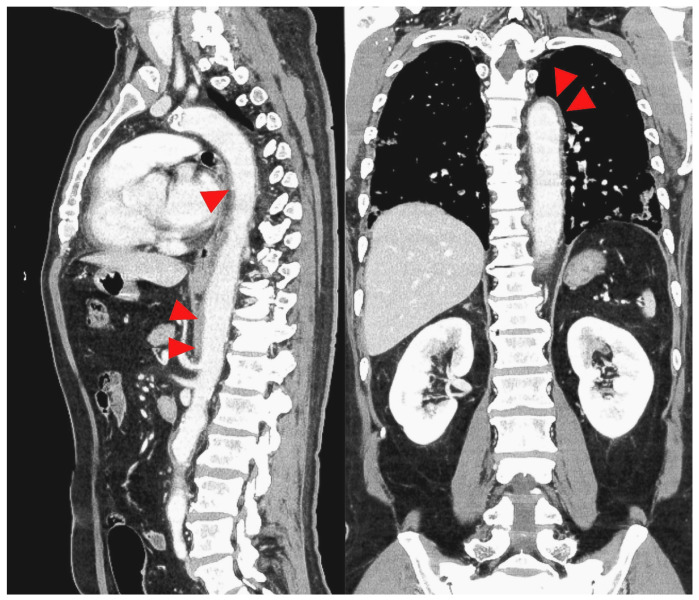
CT image with the red arrowhead indicating thickening of the thoracic to the abdominal aorta.

**Table 1 medicina-59-00816-t001:** Blood test at the time of arrival.

Test	Result	Normal Range
White blood cells (/μL)	8600	3300~8600
Red blood cells (×10^4^/μL)	4.32	4.35~5.55
Hemoglobin (g/dL)	13.4	13.7~16.8
Hematocrit (%)	40.2	40.7~50.1
Platelets (×10^4^/μL)	30.5	15.8~34.8
Total bilirubin (mg/dL)	0.4	0.4~1.5
Direct bilirubin (mg/dL)	0.16	0.05~0.30
Aspartate aminotransferase (U/L)	70	13~30
Alanine transaminase (U/L)	49	10~42
Lactate dehydrogenase (U/L)	487	124~222
Alkaline phosphatase (U/L)	79	38~113
Gamma glutamyl transpeptidase (U/L)	62	13~64
Total protein (g/dL)	6	6.6~8.1
Albumin (g/dL)	2.7	4.1~5.1
Blood urea nitrogen (mg/dL)	13	8~20
Creatinine (mg/dL)	0.69	0.65~1.07
Uric acid (mg/dL)	1.7	3.7~7.8
Sodium (mmol/L)	141	138~145
Potassium (mmol/L)	3.6	3.6~4.8
Chlorine (mmol/L)	106	101~108
C-reactive protein (mg/dL)	188.2	0.0~1.4
Procalcitonin (ng/mL)	0.04	0.00~0.05
Ferritin (ng/mL)	1720	39.9~465
Activated partial thromboplastin time	>200 s	24~39
PT-INR	1.4	0.85~1.15
Fibrinogen (mg/dL)	597	180~320
D-dimer (µg/mL)	1.88	0.0~1.0
Antithrombin III (%)	68	79~121
Krebs von den Lungen 6 (U/mL)	464	105~401
Beta-D-glucan (pg/mL)	<6.0	
Rapid plasma reagin	Negative	
Treponema pallidum hemagglutination	Negative	
IgG (mg/dL)	1131	861~1747
IgA (mg/dL)	254	93~393
IgM (mg/dL)	43	33~183

PT-INR, prothrombin time, international normalized ratio.

## Data Availability

The data that were presented in this study are available on request from the corresponding author. The data are not publicly available due to privacy restrictions.

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
