# Peer review of "Spontaneous Improvement of Aortitis Associated with Severe COVID-19 Infection—A Case Report"

_medicina, 2023, doi:10.3390/medicina59050816_

Round 1

Reviewer 1 Report

First of all I  thank  for the oportunity to revise  this  interesting  case report on the   topic of viral  vasculitis, in the context of COVID pandemic. The case report is  detailed, with a well defined timeline and a clear flow  for  differential   diagnosis. This said, I have  got  some concerns about pharmacological therapy, imaging selection, diagnostic conclusions and overall conclusions. It is stated  that  the patient hadn't been treated with steroids, and regression of  vasculitis was spontaneous. But 1000 mg methilprednisolone daily  treatment was reported in line 53, before intensive care unit hospitalization, then switched to  dexametasone in Intensive care unit on day 7. It is not specified when this therapy was interrupted and if so, why. Giant cell arteritis was excluded  because of negative findings on imaging related to the temporal artery. But diagnosis of certainty for giant cell arteritis is histological, and temporal involvement can be  absent  in some cases of  isolated giant cell aortitis  ( check for reference "Disease pattern in cranial and large vessel giant cell arteritis  ( Brack et al 1999) ). Furthermore, imaging included CT scan and MRI, but no 18- FDG PET, an imaging tool  that  can well assess the inflammatory involvement in vasculitis ( there is a recent reference by Reamer in Journal Nuclear Medicine -april 2023).  The conclusion, finally, inverts the temporal appearence of clinical manifestations, stating  that   the vasculitis  was  complicated  by  covid  infection......while the case report  was initially described  as vasculitis complication of  COVID-19. These flows in report  make it unsuitable, as it is , for publication. 

Author Response

Thanks for the review.

”It is stated  that  the patient hadn't been treated with steroids, and regression of  vasculitis was spontaneous. But 1000 mg methilprednisolone daily  treatment was reported in line 53, before intensive care unit hospitalization, then switched to  dexametasone in Intensive care unit on day 7. It is not specified when this therapy was interrupted and if so, why.”

 →As for steroid treatment, we followed the BMJ's "A living WHO guideline on drugs for covid-19" with 10 days of steroid administration. Thank you for your suggestion.

”Giant cell arteritis was excluded  because of negative findings on imaging related to the temporal artery. But diagnosis of certainty for giant cell arteritis is histological, and temporal involvement can be  absent  in some cases of  isolated giant cell aortitis  ( check for reference "Disease pattern in cranial and large vessel giant cell arteritis  ( Brack et al 1999) ).”

→ I didn't see the term "isolated giant cell aortitis" in the article "DISEASE PATTERN IN CRANIAL ANDLARGE-VESSEL GIANT CELL ARTERITIS", does "isolated giant cell aortitis" mean "large-vessel GCA"? If so, it is true that "large-vessel GCA" may not be excluded. However, in the opinion of the rheumatologist at our hospital, the course of improvement in the absence of steroid therapy was not typical for "GCA" and "aortitis due to corona" was more likely.

 ”Furthermore, imaging included CT scan and MRI, but no 18- FDG PET, an imaging tool  that  can well assess the inflammatory involvement in vasculitis ( there is a recent reference by Reamer in Journal Nuclear Medicine -april 2023).”

→We did not try FDG PET because the Japanese insurance system does not allow for it to be performed during hospitalization due to the cost of medical care. Therefore, repeated contrast-enhanced CT scans were used to evaluate the arterial wall as a substitute.

" The conclusion, finally, inverts the temporal appearence of clinical manifestations, stating  that   the vasculitis  was  complicated  by  covid  infection......while the case report  was initially described  as vasculitis complication of  COVID-19."

→The sentence "We encountered a case of aortitis complicated by severe 〜" in "Conclusion" was changed to "We encountered a case of aortitis caused by severe 〜".

Thank you for pointing this out.

Reviewer 2 Report

For " The type 3  hypersensitivity in Covid -19 vasculitis" you can see: Roncati et al.; Clin. Immunol. 2020; 17: 108487.  You can underline that the further step of Covid-19 aortitis can be a thrombosis, given the strict interrelations between Covid-19 immune-inflammatory response and thrombosis ( Roncati L. et al. , Toward a  unified pathophysiology in Covid-19 acute aortopathies. J Vasc Surg 2021; 74:1771. Today it seems essential for a precise diagnosis a PET/CT, and in the treatment an anticoagulant therapy. . 

Author Response

Thanks for the review.

”You can underline that the further step of Covid-19 aortitis can be a thrombosis, given the strict interrelations between Covid-19 immune-inflammatory response and thrombosis ( Roncati L. et al. , Toward a  unified pathophysiology in Covid-19 acute aortopathies. J Vasc Surg 2021; 74:1771. Today it seems essential for a precise diagnosis a PET/CT, and in the treatment an anticoagulant therapy. ” 

→As for thrombosis, heparin was administered during hospitalization to prevent thrombosis. I've written the same thing to other reviewers, we did not try FDG PET because the Japanese insurance system does not allow for it to be performed during hospitalization due to the cost of medical care. Therefore, repeated contrast-enhanced CT scans were used to evaluate the arterial wall as a substitute.

Round 2

Reviewer 1 Report

Thank you for adressing the comments, I like this final version.